# Empirical Dietary Patterns Associated with Food Insecurity in U.S. Cancer Survivors: NHANES 1999–2018

**DOI:** 10.3390/ijerph192114062

**Published:** 2022-10-28

**Authors:** Christian A. Maino Vieytes, Ruoqing Zhu, Francesca Gany, Amirah Burton-Obanla, Anna E. Arthur

**Affiliations:** 1Division of Nutritional Sciences, The University of Illinois at Urbana-Champaign, Urbana, IL 61801, USA; 2Department of Statistics, The University of Illinois at Urbana-Champaign, Urbana, IL 61801, USA; 3Memorial Sloan Kettering Cancer Center, New York, NY 10065, USA; 4Department of Dietetics and Nutrition, The University of Kansas Medical Center, Kansas City, KS 66160, USA

**Keywords:** cancer, dietary patterns, nutritional epidemiology, food insecurity, survivorship

## Abstract

(1) Background: Food insecurity (FI) is a public health and sociodemographic phenomenon that besets many cancer survivors in the United States. FI in cancer survivors may arise as a consequence of financial toxicity stemming from treatment costs, physical impairment, labor force egress, or a combination of those factors. To our knowledge, an understanding of the dietary intake practices of this population has not been delineated but is imperative for addressing the needs of this vulnerable population; (2) Methods: Using data from NHANES, 1999–2018, we characterized major dietary patterns in the food insecure cancer survivor population using: i. penalized logistic regression (logit) and ii. principal components analysis (PCA). We validated these patterns by examining the association of those patterns with food insecurity in the cancer population; (3) Results: Four dietary patterns were extracted with penalized logit and two with PCA. In the pattern validation phase, we found several patterns exhibited strong associations with FI. The FI, SNAP, and Household Size patterns (all extracted with penalized logit) harbored the strongest associations and there was evidence of stronger associations in those moderately removed from a cancer diagnosis (≥2 and <6 years since diagnosis); (4) Conclusions: FI may play an influential role on the dietary intake patterns of cancer survivors in the U.S. The results highlight the relevance of FI screening and monitoring for cancer survivors.

## 1. Introduction

Food insecurity (FI) is the inability to procure sufficient quantities of safe and nutritious foods that promote the physical, emotional, and psychosocial domains of health and well-being [1]. It is a pressing public health issue that affected approximately 13.8 million (10.5%) U.S. households in 2020 and disproportionately implicates low-income households, single-parent households, communities of color, and those with a recent diagnosis of cancer [1,2]. For many households, experiencing a sudden cancer diagnosis and its side effects may worsen FI status. Increasing treatment costs and side effects attributable to those treatments may prompt lower quality of life (QOL) and physical disability in cancer survivors, which magnify the risks of unemployment and financial sequelae [3,4]. The culmination of these and other known risk factors of FI, including younger age, being less educated, belonging to a marginalized community, and having lower income, may ultimately lead to cancer survivors experiencing FI [5]. Moreover, estimates from non-nationally representative data suggest that the prevalence of FI in the cancer survivor (defined as any person with a history of cancer, from the time of diagnosis to the end of life) population may be higher than the national average and it is unclear if time elapsed since diagnosis plays a role in this phenomenon [6,7,8].

National guidelines from the WCRF/AICR Third Expert Report have developed recommendations that cancer survivors may implement following a diagnosis. These recommendations include dietary modifications that emphasize the consumption of whole grains, vegetables, and fruit while curtailing the consumption of sugar-sweetened beverages and processed meats, as higher intakes of these foods may be associated with an increased cancer risk and worsen the prognosis [9]. Though following these evidence-based guidelines may improve QOL and disease outcomes, it is unclear how FI impacts cancer survivors’ capacity to adhere to those recommendations [10,11,12,13]. Ultimately, the combination of treatment-associated sequelae and FI may aggravate nutritional inadequacy in food insecure cancer survivors.

Ascertaining population-specific dietary patterns may reveal critical needs and play a role in developing clinical best practices or food policy targeted at specific at-risk populations. Consequently, the goal of this study was to delineate major dietary intake patterns among food insecure cancer survivors by implementing dietary pattern extraction procedures on nationally representative data from the National Health and Nutrition Examination Survey (NHANES). We implement penalized logistic regression, a novel methodology for dietary patterns analysis embraced by colleagues, and principal components analysis (PCA) to empirically characterize the dietary patterns of our target population [14,15]. We subsequently validate those patterns by examining their relationship to the risk of being food insecure. To our knowledge, there are no studies evaluating the dietary patterns of food insecure cancer survivors using nationally representative data and this is the first study to employ NHANES data to analyze empirical dietary intake patterns in cancer survivors with self-reported FI.

## 2. Materials and Methods

Data from ten consecutive cross sections of the NHANES study, between 1999–2018, were employed for the analysis. The analytical outline and strategy are displayed in Figure 1. NHANES is a biennial national cross-sectional study conducted by the Center for Disease Control and Prevention (CDC) and the National Center for Health Statistics (NCHS), that surveys health, nutrition, and other lifestyle factors across the noninstitutionalized civilian population of the United States [16]. The study employs a multistage probability selection design to generate a nationally representative sample of the American population and to ascertain the prevalence of diseases, health outcomes, and associated environmental and behavioral risk factors [17]. Consenting participants fulfill a household screener and a home interview. The latter consists of a series of questionnaires administered in their homes that cover a range of areas, including demographic, occupational, health, and dietary-related matters. Some subjects are selected for a medical examination, which includes a variety of physical measurements, a dental examination, and biological specimens for laboratory testing. Examination data were collected in the Mobile Examination Center. In addition, dietary data were collected via 24 h recalls to ascertain the frequency and quantity of dietary consumption and estimate nutrient intake. Cancer, diabetes, cardiovascular disease, and renal disease statuses are assessed as self-reported items in the Medical Conditions Questionnaire (MCQ). Tumor stage data for cancer survivors are not part of the survey. All study procedures and protocols were approved by the NCHS Ethics Review Board and all participants provided informed and written consent [17].

### 2.1. Study Sample

Figure 1 details a flow diagram of the sample selection process. We used data from ten survey cycles between 1999–2018 that included a subsample of 5166 participants aged at least 20 years with a self-reported history of cancer and reliable dietary data, as defined by the NCHS. Cancer status and history were ascertained in the MCQ by asking “Have you ever been told by a doctor or health professional that you had cancer or a malignancy of any kind?”. Individuals reporting a history of non-melanoma skin cancer (*n* = 576) and no other cancer type were recoded as not having a significant cancer history given the generally benign course associated with these cancers that might otherwise bias the sample [18]. Dietary patterns extraction procedures using penalized logistic regression models were performed on individuals reporting a history of a cancer diagnosis and who demonstrated complete records for food security status, data on receipt of Supplemental Nutrition Assistance Program (SNAP) benefits, household size, and age (subsample A, *n* = 3317). To extract dietary patterns that characterized intake in the population of food insecure cancer survivors using PCA we further excluded individuals reporting full or marginal food security (*n* = 2884) (i.e., only food insecure cancer survivors—subsample B, *n* = 433). Validation analyses examining the relationship between computed diet pattern indices and the risk of FI were performed on the pooled subsample of food secure and food insecure cancer survivors (subsample A, *n =* 3317).

### 2.2. Demographic and Physical Health Covariates

Demographic characteristics were self-reported and captured in the home interview. Age was modeled continuously, and sex was coded dichotomously (Male and Female). Race and ethnicity were categorized as Mexican American, Other Hispanic, Non-Hispanic White, Non-Hispanic Black, and Other/Multiracial (although we note that our final analytical models implemented a binary-coded version given the small sample size—non-Hispanic White and other). We considered income status using the family income-to-poverty ratio (FIPR), classified into two categories: <1.3 or ≥1.3. This value was chosen deliberately as it is a threshold commonly employed by various federal safety net programs to evaluate low socioeconomic status for program eligibility [19]. We modeled household size numerically.

Health-related and behavioral characteristics included body mass index (BMI) (modeled continuously in units of kg/m^2^), smoking status, which was categorized as current smoker (currently smoking every day or some days), former smoker (not currently smoking but with a lifetime history of ≥100 cigarettes), or never smoker (a lifetime history of smoking <100 cigarettes), and drinking status, which classified participants as heavy drinkers (≥14 g/d for women and ≥28 g/d for men), moderate drinkers (0.1–13.9 g/d for women and 0.1–27.9 g/d for men), and abstainers (<0.1 g/d)—note: 1 serving of alcohol equates to 14 g of alcohol [20,21]. Finally, we computed a modified version (given limitations with the data provided through NHANES and those required for full computation of the metric) of the Charlson Comorbidity Index and weekly metabolic equivalents (MET), as previously described, to evaluate comorbidity burden and physical activity (measured as all physical activity exerted on a weekly basis), respectively [22,23]. In subsequent modeling efforts, these measures were modeled as continuous variables.

### 2.3. Dietary Assessment Data

Dietary data were collected using the 24 h recall method from NHANES participants during an in-person interview (performed in the MEC) [24]. A subsequent, unannounced 24 h recall is collected via telephone within 3–10 days following the interview. The dietary interview protocols and the administered 24 h recall were designed to provide detailed dietary data by capturing the foods and beverages consumed by participants within the preceding 24 h. The methodology for the dietary interview component was developed by the USDA’s Food Surveys Research Group and incorporated the USDA’s automated multiple-pass method [24,25]. Dietary data collected between 1999 and 2002 included only one day of intake from participants, whereas data collected between 2003 and 2018 included two days of recalls from each participant. To make full use of the available data and minimize any bias introduced by using a single day of dietary intake values, we averaged intake values across both days of data collection [26,27]. Daily total energy and nutrient intake data were obtained for each participant. Total energy and nutrient intake values were estimated from foods noted in the dietary interview while cross-referencing the Food and Nutrient Database for Dietary Studies [28].

Intake according to food groups data were obtained from the publicly available USDA Food Patterns Equivalents Database (FPED) and MyPyramid Equivalents Database (MPED) [29,30]. The FPED and MPED use a database of 8356 commonly consumed food items to compute intake equivalents across 37 food pattern components. Considering this classification scheme, a modified yet similar, food-grouping scheme involving 26 food groups was adopted for this analysis. These 26 groups and the way they were collapsed are detailed in Table (Appendix A). Prior to any dietary patterns extraction procedures, food group intake equivalents were divided by a subject’s total caloric intake so that a multivariate density model could be implemented to adjust for total energy intake and minimize the likelihood of confounding by total energy intake in any of the subsequent modeling efforts [31].

### 2.4. Cancer Status and Food Security Data

The MCQ provides survey participants with an avenue for self-reporting data on medical conditions. Time since cancer diagnosis was computed as the time elapsed between a subject’s age at their first cancer diagnosis and their current age and was subsequently categorized (<2 years, ≥2 and <6 years, and ≥6 years). Participants with a history of a cancer diagnosis were grouped into their primary cancer type. That is the cancer type with the longest associated time since diagnosis. Lastly, the 32 cancer types listed in the NHANES MCQ were collapsed into a set of 8 primary cancer groups proposed by colleagues (Breast, Gastrointestinal, Genitourinary, Gynecological, Male Reproductive, Melanoma, Skin-Unknown, and Other) [32]. Again, individuals reporting only a diagnosis of non-melanoma skin cancer and no other cancer were omitted as positive cancer cases, given the generally benign course associated with this malignancy [18].

Food security status was assessed using the U.S. Food Security Survey Module (U.S. FSSM), an 18-item screener employed by NHANES since the 1999 cycle to assess food security experienced by subjects over the preceding year [33,34]. The questionnaire was administered in the home interview setting, with one adult responding on behalf of all individuals in that household, regardless of whether they were included in the survey. The survey is comprised of 10 items dedicated to households without children and eight items for households with children. Counts and affirmative responses on the questionnaire are used to bin subjects into overall food security categorizations. Those responding in the affirmative to ≤2 items were categorized as food secure, while those responding in the affirmative to ≥3 items were categorized as food insecure, and followed validated cutoffs [33,34]. Additionally, receipt of food assistance and specifically participation in the Supplemental Nutrition Assistance Program (SNAP) is reported in the U.S. FSSM. These data were captured by prompting participants on whether any household member was authorized to receive SNAP benefits in the 12 months preceding the interview.

### 2.5. Statistical Analysis

Descriptive statistics were tabulated on demographic variables across levels of food security/cancer status using subsample A (Figure 1). Dietary patterns were extracted using penalized logistic regression models and PCA (see Appendix B for a detailed description of the implementation of these procedures). For the former, we used four binary outcomes: food insecurity status (food insecure vs. food secure), age ≥60 years, household receipt of SNAP benefits in the last 12 months, and household size ≥5, which are all associated or understood risk factors for FI [35,36]. A Pearson correlation matrix was generated to evaluate relationships between the dietary patterns and food groups in subsample A (*n* = 3117). To validate the extracted dietary patterns, we used the loadings and coefficients (from the PCA and elastic net procedures, respectively) to compute dietary pattern scores for subjects identifying with a history of cancer (subsample A, *n* = 3317). The validation phase of the analysis comprised the analytical goal of determining the relationship between the extracted pattern scores and the risk of FI in the cancer survivor population (Figure 1). To this end, we implemented weighted logistic regression models that modeled the log odds of being food insecure as a function of the dietary patterns scores and relevant covariates. This step included all subjects with a reported history of cancer (subsample A, *n* = 3317). Alcohol consumption was not included as a covariate in these models to minimize collinearity, given that the extracted patterns already considered alcohol consumption in their computation.

We modeled the diet scores using several specifications to evaluate the robustness of the results. First, we modeled the scores categorically after binning participants into quintiles. A test for linear trend across the quintiles was performed by generating a new variable that assigned each subject the median value of their respective quintile and modeling it as a continuous variable. Second, we standardized the diet scores by dividing them by their respective standard deviation and then modeling them as continuous variables. Third, we added a quadratic term to the previous model to assess for divergence from a linear relationship. Lastly, we modeled the diet scores using restricted cubic splines with five knots to model the relationship flexibly and evaluate for dose–response and, again, linearity. All models were adjusted for relevant confounders including age, sex, race/ethnicity, family income-to-poverty ratio, highest level of education attained, household size, SNAP participation status, BMI, estimated caloric intake, weekly MET minutes, primary cancer site, smoking status, and the Charlson Comorbidity Index score. We fit stratified models according to sex, time since primary cancer diagnosis, and attained level of education. We accounted for the complex and multi-stage probability design of the study by following NCHS analytical guidelines and weighting our analyses accordingly [37]. All analyses were conducted at α = 0.05 and were performed in R version 4.2.1 (The R Foundation, Vienna, Austria). All accompanying R code and data files necessary to reproduce these analyses can be accessed on GitHub at: https://github.com/cmainov/NHANES-Diet-Penalized-Regression.

## 3. Results

### 3.1. Descriptive Statistics

Sociodemographic, clinical, and behavioral characteristics are summarized in Table 1. On average, those with a reported history of cancer and with self-identified low food security were younger than food secure cancer survivors, were more likely to identify as female, live in a home with ≥5 individuals, and belong to a minority group compared to those identifying as food secure with a history of cancer who were older, had a more balanced ratio of the sexes, and were predominantly white. Food insecure survivors also had lower attained educational status compared to food secure individuals, tended to have a lower FIPR, reported being more physically active throughout the week, and were more likely to be receiving food assistance through SNAP in addition to, on average, consuming over 200 fewer daily calories compared to their food secure counterparts. Regarding cancer site, food insecure survivors disproportionally reported gynecological cancers as their primary form of cancer compared to a lower rate in the food secure survivors. Concomitantly, a larger proportion of male reproductive cancers were represented in the food secure sample relative to the food insecure sample. There was no gross difference observed in time-since-diagnosis across the two groups. However, food insecure individuals had a slightly larger mean Charlson Comorbidity Index score than food secure participants. Finally, those identifying as food insecure were more likely to report being current smokers than food secure individuals with cancer.

### 3.2. Discovery Phase: Dietary Patterns Extraction

There were six dietary patterns extracted from both procedures. The patterns derived using penalized logistic regression were named according to the outcome variable used in each model (we named these the Food Insecurity (FI), Age, SNAP, and Household Size patterns, respectively). Figure 2 illustrates the optimal combinations of the model tuning parameters, λ and α, that were ultimately selected for each model. For the model with FI as the response variable, the LASSO regression (α=1) solution was optimal while the ridge regression solution (α=0) was optimal for the model with household size as the response. The models with age and SNAP benefits as the outcomes yielded optimized solutions with α in the elastic net range, α∈ (0,1). The coefficients for each of these models are found in Appendix A. We note that the coefficients for several food groups shrunk to zero, effectively eliminating them from subsequent score computations.

In Table 2, we detail the Pearson correlation coefficients amongst pattern scores and food groups. We used a cut-off threshold of |0.30| to identify food groups that significantly contributed to these patterns [38]. The FI pattern was positively correlated with intakes of potatoes, and added sugars while negatively correlated with intake of other vegetables. The Age pattern was positively correlated with intakes of milk, fruit, and whole grains while negatively correlated with cheese. Overall, this pattern was negatively correlated with the FI pattern (*r* = −0.28). The SNAP dietary pattern was strongly and positively correlated with the FI pattern (*r* = 0.80) as well as with added sugars while being negatively correlated with alcohol, dark-yellow vegetables, other vegetables, and nuts. The final Household Size pattern was also strongly and positively correlated with the FI pattern (*r* = 0.63) and negatively correlated with intakes of yogurt, other fruit, citrus, melons and berries, tomatoes, and other vegetables being positively correlated with intake of added sugars.

For the patterns extracted with PCA, we evaluated a scree plot initially and found that an “elbow” appeared after the fourth principal component (Appendix A). However, upon evaluation of the component loading matrix (Appendix A) and the table of correlations (Table 2), only the first and second principal components had interpretable loadings that were deemed meaningful. Thus, a decision was made to retain only the first two components given the weight placed on having interpretable components [39]. The eigenvalues suggested that these first two components accounted for 14.1% of the variation in the 24 h recall data. Both patterns shared similarities in that both were positively correlated with vegetable consumption and negatively correlated with added sugar and alcohol. The first principal component emphasized oils, cheese, tomatoes, and other vegetables and it negatively emphasized added sugars. It also modestly emphasized consumption of processed meat, meats, cheese, solid fat, milk, and eggs. The second principal component was also negatively correlated with solid fats and refined grains intakes and positively correlated to intakes of poultry, high n-3 seafood, yogurt, other fruit, citrus, melons, and berries, dark-green vegetables, dark-yellow vegetables, and other vegetables. Given both healthful and unhealthful aspects of the first principal component, we termed this pattern the Modified Western pattern [40,41]. In contrast, the second principal component was termed the Prudent pattern, given its greater and more consistent emphasis on the pillars of healthful eating cited previously in the literature [42].

Differences across sociodemographic covariates between high and low median splits of each of the six dietary patterns in the subsample of cancer survivors (subsample A, *n* = 3317) are presented in Table 3. On average, those with higher scores on the Age pattern tended to be older. Subjects with greater scores on the FI, SNAP, and Household Size patterns also tended to be younger and have a lower FIPR than those with lower pattern scores. Subjects with high scores on the household size pattern were also more likely to report living in a home with ≥5 persons compared to low scorers, while high scorers on the FI and SNAP patterns were more likely to identify as food insecure and receive SNAP benefits compared to low scorers. Finally, high scorers on the Prudent pattern were, on average, more likely to report as never smokers and less likely to report as current smokers compared to low scorers.

### 3.3. Validation Phase: Logistic Regression

Using binary logistic regression models, we found, after multivariable adjustment, significant associations between the extracted pattern scores and the odds of being food insecure (Table 4). The FI, SNAP, and Household Size patterns were all strongly and positively associated with the risk of being food insecure. Among those, the FI pattern had the most considerable magnitude of association, with the odds of FI being 2.42-fold greater in the fifth quintile compared to the first quintile. Similarly, all three patterns had similar magnitudes of association when the diet scores were modeled continuously. For the FI pattern, a one standard deviation increase in the score was associated with 1.50-fold increase in the odds of being food insecure. Concerning associations in the opposite direction, only the Prudent pattern was inversely associated with FI, with the highest quintile observing a 60% reduction in the odds of being food insecure compared to the first quintile. A one standard deviation increase in this pattern scores was also significantly associated with a 24% decrease in the odds of being food insecure. For all the noted dietary patterns, tests for linear trends revealed linear behavior, in their respective directions, across the quintiles, and these findings were generally supported by the results from fitting models with restricted cubic splines (Figure 3), although the strongest relationship, again, appeared to belong to the FI pattern.

Results from stratified models are presented in Appendix A. We found that relationships between each diet pattern score and FI risk were significantly stronger in females than males. The highest quintile of the FI pattern demonstrated a 3.48-fold greater risk of being food insecure compared to the lowest quintile, while the signal in the male population was blunted with only a non-significant 1.46-fold greater risk of FI in the fifth relative to the lowest quintile. Likewise, comparisons between the SNAP, Household Size, and Prudent patterns displayed similar phenomena (Appendix A). When comparing time since primary cancer diagnosis, we found that the FI pattern was associated with 4.72-fold greater risk of FI in those subjects intermediately removed from a cancer diagnosis at greater than two and less than six years removed from their diagnosis. Within this group, the strongest association belonged to the SNAP pattern, where the fifth quintile demonstrated a 7.90-fold greater risk of FI than the lowest quintile, and a one standard deviation increase in pattern score was associated with a significant 1.65-fold increased risk of FI. In those ≥6 years removed from their primary diagnosis, there was a positive relationship between greater adherence to the FI pattern and a greater risk of FI. Moreover, the highest quintile of the prudent pattern was associated with a 67% reduction in the risk of being food insecure compared to the first quintile and there was evidence of a significant linear trend. Finally, when examining education status, it was revealed that strong and significant associations were present in the FI, SNAP, Household Size, and prudent patterns for those reporting some level of college or greater but not those with only a high school education or less.

## 4. Discussion

The results we present highlight major dietary patterns associated with FI in the cancer survivor population, a population plagued by high nutritional requirements and, often because of treatment-related or other side effects, limited nutritional intake. Using a combination of empirical methods, we extracted six dietary patterns to characterize the dietary intake patterns of this population. We used supervised learning in the form of penalized logistic regression to model FI and other risk factors of FI by regressing them on the 26 food groups considered in the analysis. Some of the resulting patterns were similar and consistent in that three emphasized comparable food groups, all to a similar extent, although they contained notable differences. Namely, high consumption of added sugars and low consumption of various classes of whole fruits and vegetables were themes consistently seen in those patterns (FI, SNAP, and Household Size patterns). Decreased consumption of whole grains, nuts, and legumes also highlighted these patterns, which, taken together, may suggest that food insecure survivors were, on average, more likely to be following a diet comprised, principally, of processed foods. Regarding their relationship to FI in the cancer survivor population, we found that the FI, SNAP, and Household Size patterns, in particular, were strongly and positively associated with the risk of FI.

Within the broader context of studies addressing FI and diet quality, we found that those patterns extracted from the data using penalized logistic regression shared many similarities with similar studies done in other populations. In particular, the strong relationship between the extracted patterns and intake of added sugars is consistent with reports stressing the pervasiveness of sugar-sweetened beverages and added sugar consumption in other food insecure populations [43,44,45,46,47]. Furthermore, though several classical studies employing unsupervised learning methods such as PCA to extract dietary patterns empirically have consistently yielded “Western-style” patterns that highlight high consumption of meat and processed meat, this was not a consistent finding in our study. Meat intake was emphasized to a modest degree in the patterns we extracted with penalized logistic regression. Nonetheless, this finding, supported by evidence elsewhere in the literature, may highlight restraint on the part of food insecure individuals from purchasing more cost-prohibitive food items, such as meat, and resorting to other low-cost and high-calorie alternatives instead [48]. All in all, we find that the clinically meaningful evidence we describe lends further support and validation of the penalized logistic regression approach as a viable alternative for extracting dietary patterns that are outcome specific.

As a comparative analysis, we implemented unsupervised learning in the form of PCA to derive dietary patterns. PCA is a powerful tool that is an established method of deriving dietary patterns but also suffers from limitations. For instance, the interpretability of the principal components may be equivocal [49]. Moreover, PCA may not always be a suitable approach for extracting patterns associated with a condition or disease outcome. These notions were substantiated by the results of our study as well. Given that the procedure only aims to constrain as much of the variation in the dietary intake data onto a single dimension, predictive potential is not a guaranteed result when implementing this approach [49]. We found that the patterns extracted using penalized regression were more consistent with previous reports in the literature detailing diet quality in other food insecure subpopulations. Moreover, considering the supporting studies we describe above, we stress that in our study, PCA did not yield dietary patterns consistent with diet quality patterns described in other food insecure populations.

In the validation phase of our analysis, we found that the FI, SNAP, and Household Size patterns were positively and strongly associated with FI, while the Prudent pattern was strongly and inversely associated with being food insecure in the cancer survivor population after controlling for several relevant confounders. Stratified analyses yielded peculiar findings. Notably, we found that stratifying the validation models by sex revealed strong effect sizes for the aforementioned association in females but not males. This finding was particularly interesting when evaluated in the context of preceding studies reporting sex-specific disparities within FI research. FI has been demonstrated to be a highly gendered and sex-specific outcome that disproportionately affects females and, specifically, females that head households as opposed to males in male-headed households [50]. Downstream of FI itself, it is also understood that FI impacts males and females disparately concerning clinical outcomes, with food insecure females being significantly more likely to experience obesity compared to their food insecure male counterparts [51,52,53,54,55]. A biological basis for explaining these disparate associations is not readily accessible, with some in the field suggesting that these relationships may be explained by the gendered societal norms concerning childcare traditionally imposed on women [51]. We posit that our results may provide an additional layer of evidence for understanding the dynamic between FI and sex-specific disparities, though we are limited in our conclusions given the potential for reverse causality due to the cross-sectional design. Nevertheless, within the context of the food insecure cancer population, this conjecture would only help to clarify those relationships in younger cancer survivors of child-bearing age and not necessarily post-menopausal survivors.

In addition to strong effect sizes in females, there were disparities across time since primary diagnosis that emerged. The association between the penalized logistic regression patterns and FI was strongest in those 2–6 years removed from a cancer diagnosis. These findings suggest that dietary intake may be more relevant for predicting FI status among cancer survivors within this group and not necessarily those proximal or distal to a diagnosis. However, previous findings in the literature have not found a significant moderating effect of time since diagnosis on FI-related outcomes [8,56]. Therefore, it is challenging to clarify how time since diagnosis may be moderating the results in our validation models. Nonetheless, our results may be consistent with the hypothesis that any FI resulting from financial hardship encountered throughout the cancer care continuum may not impact survivors immediately and may persist for several years before abating. Though the results have been mixed among different studies, we believe that this is an area that requires further scrutiny if we are to understand the dynamics of food insecurity throughout the cancer care continuum.

The results we present have global public health ramifications. Clinically, FI continues to be an underappreciated social determinant of health, mainly afflicting low-income populations. A consequence of FI manifests in the trade-offs exacted on food insecure survivors when faced with competing demands of nutrition and medical care [57]. Furthermore, there are currently no known recommendations or guidelines from any influential medical association or organization stressing the need for food security screenings in this population, again underscoring the urgency and relevance of this research [57]. It was previously shown that the rates of FI in the cancer population may be substantial in the low-income cancer population compared to the general population [58]. Social and economic factors are especially crucial in prognosis and survival following diagnosis, and nutrition may be a mediating factor in survivorship. Moreover, it is imperative to underscore that compared to food secure cancer patients, food insecure cancer patients comprised a substantially larger proportion of individuals from minority racial and ethnic groups, which is also consistent with what has previously been reported [59]. Though this analysis was completed with data from the U.S., we believe that many of these findings and considerations are germane in a global context. In particular, these findings may be relevant for clinicians and cancer survivors in countries without universally subsidized health care, like the U.S. Nevertheless, many facets of cancer survivorship, such as job loss and physical disability, still define the QOL cancer survivors experience globally and remain as factors that may impede access to healthy and nutritious food. Finally, this work utilizing penalized logistic regression also corroborates a novel and pre-existing framework for evaluating dietary patterns associated with particular exposures [14,15]. This approach may be helpful not only for evaluating the dietary patterns of specific populations, as we have demonstrated here but also for monitoring and evaluating the effects of nutrition policy initiatives in the U.S. and abroad.

Considering the study’s findings within the framework of guidelines established in the WCRF/AICR third expert report, we conclude that the cancer food insecure population within the United States may be hindered from meeting the report’s benchmarks. The report stresses the vital role of fruits, vegetables, legumes, and whole grains in the prevention of incident cancer, cancer control, and bolstered survivorship [9]. We found that dietary patterns derived in both manners, although more robustly with penalized logistic regression, suggested that FI in the cancer survivor population was associated with poor diet quality that was not aligned with those guidelines. Future studies, specifically those with longitudinal cohort designs, should more closely examine the relationships between FI and dietary intake with prognostic and other functional outcomes in this population. Nevertheless, this research elevates the importance of utilizing the WCRF/AICR guidelines in clinical settings and, in particular, subsequent to food insecurity screenings.

This analysis has several strengths, including the large, combined sample size, nationally representative sampling, control for other confounding variables, and the use of a validated module for measuring food security status. There are weaknesses in our study worth noting. As is characteristic in observational studies, residual confounding and the presence of reverse causality cannot be ruled out, particularly given the cross-sectional study design. Whether FI caused the observed dietary patterns or vice versa is not a conjecture we can explicitly arrive at with these data. Furthermore, the use of a household FI metric is essential to consider, given that FI at the household level may impart unequal burdens on its residents and, in addition, the fact that these data are self-reported in nature. With regard to dietary intake measurements, we cannot rule out any systematic biases introduced by the dietary measurement protocol. In this vein, using a 24 h recall instead of a more robust measure of dietary intake, such as a food-frequency questionnaire, is also a notable limitation. Moreover, we must again stress that there are, to our knowledge, no current published design-based modeling software allowing users to perform penalized regression (e.g., Ridge or LASSO regression) on complex survey data. Nonetheless, as we did in our analysis, weighting those procedures with normalized weights (see Appendix B) was a deliberate strategy for curtailing any parameter or standard error bias introduced by not using all components of the complex survey design [60,61]. Concerning the use of a posteriori methods for dietary patterns extraction, we also concede that these methods are limited in that they do not allow us to make explicit recommendations on absolute values of dietary intake for any given food group analyzed, in contrast to some a priori diet quality indices. Finally, we must also consider that selection bias may arise when we include a greater proportion of individuals further removed from their diagnosis that may have less aggressive or more treatable forms of cancer that may preclude them from impaired eating, experiencing debilitating cachexia, or otherwise worse prognoses.

## 5. Conclusions

In summary, we conclude that dietary intake in the food insecure cancer population may be nutritionally inadequate, as measured by guidelines from numerous national institutions and organizations, and is characterized by consumption of processed and unhealthful foods with a concomitant dearth of fruits and vegetables [9,21]. These deficiencies are essential to highlight in a nutritionally vulnerable population already suspectable to malnutrition as they may lend themselves to poorer clinical outcomes, though further evidence is warranted. In addition to evaluating the effects of these dietary patterns on clinical outcomes, future studies, particularly prospective longitudinal cohort studies, are needed to highlight the impact that nutritional consequences of FI have on cancer-related outcomes. Ultimately, the results of this analysis reinforce the notion of food security as a critical social determinant of health with consequences to nutritional intake that may require persistent screenings. These findings are critical and impactful given that there are currently no best-practice guidelines or consensus criteria within the cancer survivor population to ultimately abrogate the prevalence of FI and bolster patient prognoses [57].

## Figures and Tables

**Figure 1 ijerph-19-14062-f001:**
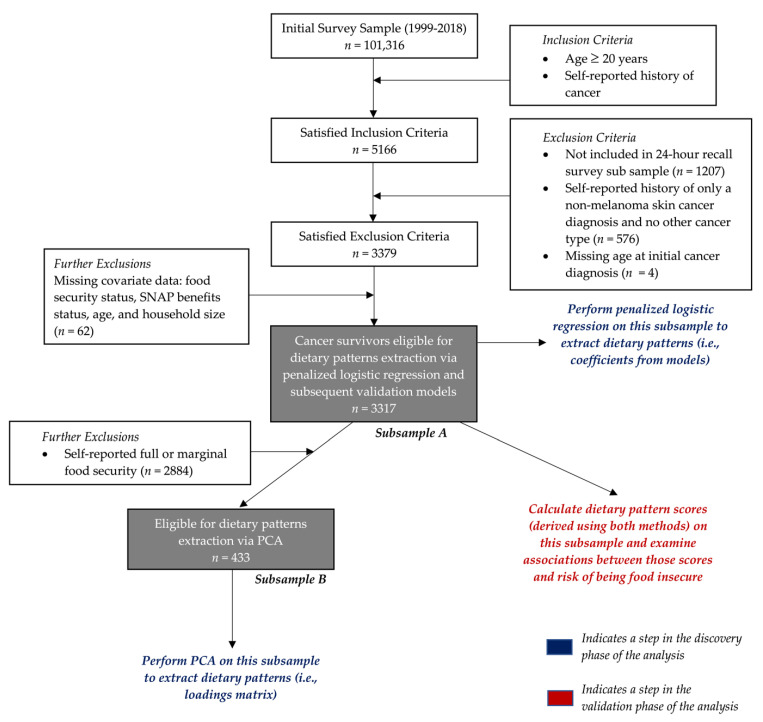
Study sample flow chart detailing the sample selection process and the analytical strategy. Subsamples A and B are periodically referred to in the text.

**Figure 2 ijerph-19-14062-f002:**
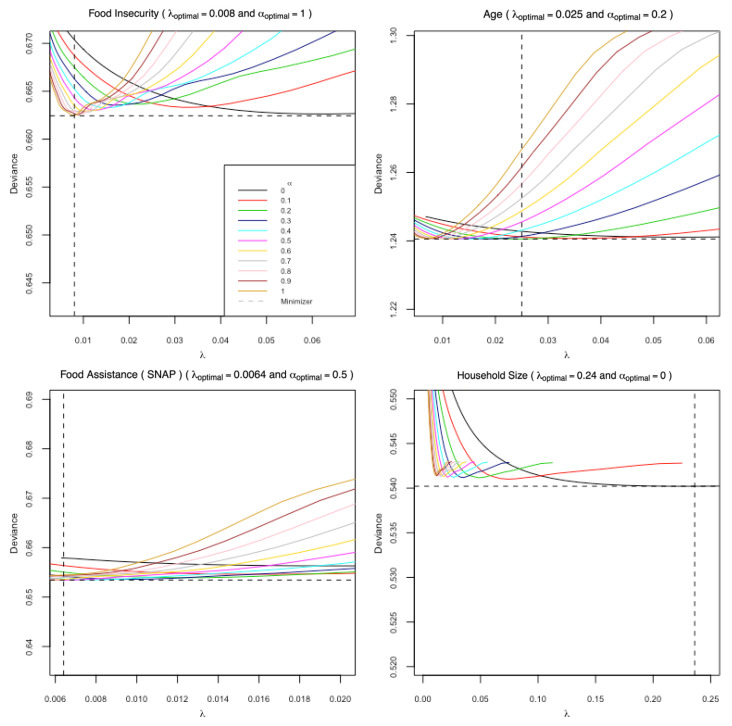
Optimal Combinations of α and λ (minimizers) in the penalized logistic regression models used for dietary patterns extraction performed on subsample A (*n* = 3117).

**Figure 3 ijerph-19-14062-f003:**
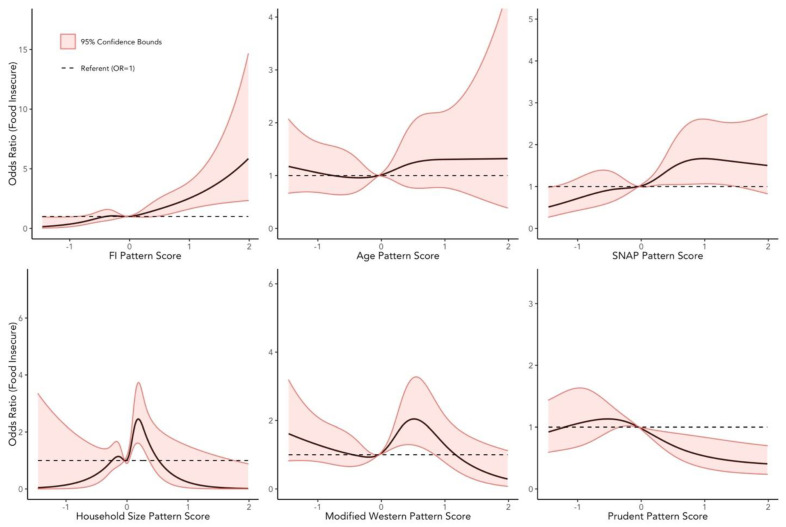
Adjusted restricted cubic spline curves demonstrating the relationships between dietary pattern scores and the odds of being food insecure in the subsample of cancer survivors (subsample A in Figure 1). These models used five knots to model each dietary pattern score and adjusted for age, sex, race/ethnicity, income to poverty ratio, highest level of education attained, household size, SNAP participation status, BMI, estimated caloric intake, weekly MET minutes, smoking status, and the Charlson Comorbidity Index (CCI) score and were weighted using normalized weights. The hazard at the median of each dietary pattern score was employed as the referent.

**Table 1 ijerph-19-14062-t001:** Sociodemographic and behavioral characteristics of the cancer survivor study sample (subsample A in Figure 1) and stratified by food security status. Frequencies and means are presented with percentages and standard deviations, respectively, in parentheses.

Characteristic	Total Survivors (*n* = 3317)	Food Insecure (*n* = 433)	Food Secure (*n* = 2884)	*p*
Age				<0.01
Mean (SD)	62.6 (14.8)	50.8 (15.7)	64.1 (14)	
Sex				<0.01
Male	1527 (40.9)	143 (24.5)	1384 (42.9)	
Female	1790 (59.1)	290 (75.5)	1500 (57.1)	
Race/Ethnicity				<0.01
Mexican American	235 (2.7)	67 (9.2)	168 (1.9)	
Other Hispanic	183 (2.6)	48 (5.8)	135 (2.2)	
Non-Hispanic White	2219 (84.3)	208 (68.3)	2011 (86.3)	
Non-Hispanic Black	534 (6.9)	88 (11.2)	446 (6.4)	
Other/Multiracial	146 (3.5)	22 (5.5)	124 (3.2)	
Education Attained				<0.01
≤High School	1577 (36.8)	279 (59.3)	1298 (34.1)	
≥Some College	1737 (63.2)	152 (40.7)	1585 (65.9)	
FIPR				<0.01
≥1.3	2279 (82.2)	128 (39.1)	2151 (87.6)	
<1.3	800 (17.8)	288 (60.9)	512 (12.4)	
Household Size				<0.01
<5 Persons	3027 (92.3)	345 (79.3)	2682 (93.9)	
≥5 Persons	290 (7.7)	88 (20.7)	202 (6.1)	
BMI (kg/m^2^)				0.23
Mean (SD)	29.2 (6.6)	29.7 (7.2)	29.1 (6.5)	
Weekly MET Minutes				<0.01
Mean (SD)	2314.2 (4475.2)	4641.1 (7771)	2034.9 (3804.1)	
Daily Caloric Intake (kcal)				<0.01
Mean (SD)	1894.6 (687)	1711.1 (740.2)	1917 (677.0)	
Charlson Comorbidity Score				<0.01
Mean (SD)	3.0 (1.4)	3.3 (1.8)	3.0 (1.4)	
SNAP Assistance				<0.01
No	2839 (88.6)	220 (49.9)	2619 (93.3)	
Yes	478 (11.4)	213 (50.1)	265 (6.7)	
Cancer Site				<0.01
Breast	563 (17.2)	58 (10.2)	505 (18.0)	
Gastrointestinal	321 (7.7)	45 (11.3)	276 (7.3)	
Genitourinary	145 (3.7)	15 (3.7)	130 (3.7)	
Gynecological	522 (17.8)	132 (38.1)	390 (15.3)	
Male Reproductive	620 (13.8)	50 (6.0)	570 (14.7)	
Melanoma	240 (9.3)	15 (2.0)	225 (10.2)	
Other	592 (19.1)	99 (23.1)	493 (18.6)	
Years Since Diagnosis				0.53
<2 years	817 (22.0)	113 (21.4)	704 (22)	
≥2 and <6 years	1991 (64.4)	257 (67.4)	1734 (64.1)	
≥6 years	497 (13.6)	60 (11.2)	437 (13.9)	
Smoking Status				<0.01
Current	517 (16.4)	142 (37.8)	375 (13.8)	
Former	1347 (38.9)	120 (26.8)	1227 (40.4)	
Never	1451 (44.7)	170 (35.4)	1281 (45.8)	
Alcohol Use				0.13
Heavy	323 (12.8)	29 (6.6)	294 (13.6)	
Moderate	498 (16.1)	48 (15.0)	450 (16.2)	
None-drinking	2496 (71.1)	356 (78.5)	2140 (70.2)	

Percentages may not add to 100% given rounding; *p*-values are from chi-square tests for categorical variables and *t*-tests for continuous variables.

**Table 2 ijerph-19-14062-t002:** Pearson correlation coefficients matrix amongst each of the derived patterns (extracted using either penalized logistic regression or principal components analysis) and the food groups used in the analysis. A lower triangular Pearson correlation matrix is appended, showing the correlations between the extracted dietary patterns. This analysis was performed on subsample A (Figure 1), an analytical subsample of cancer survivors (*n* = 3317).

Pattern	Food Insecurity (FI) ^†^	Age ^†^	Food Assistance (SNAP) ^†^	Household Size ^†^	Modified Western ^‡^	Prudent ^‡^
Food Groups						
Processed Meats	−0.05	−0.01	0.04	0.03	0.12	−0.22
Meats	0.22	−0.03	0.08	0.00	0.07	−0.17
Poultry	0.00	−0.26	−0.08	0.20	−0.03	**0.35**
Seafood—High n-3	−0.16	0.05	−0.11	−0.06	−0.04	**0.30**
Seafood—Low n-3	−0.17	0.08	−0.06	−0.16	−0.04	0.07
Eggs	0.07	0.11	0.00	0.15	0.24	0.07
Solid Fats	0.12	0.04	0.22	−0.12	0.21	**−0.46**
Oils	−0.26	0.04	−0.24	0.09	**0.34**	0.12
Milk	−0.10	**0.37**	0.00	−0.19	−0.06	0.12
Yogurt	−0.09	0.05	−0.10	**−0.32**	0.07	**0.35**
Cheese	−0.06	**−0.39**	0.05	−0.19	**0.34**	−0.28
Alcohol	−0.19	−0.27	**−0.34**	−0.09	**−0.36**	−0.16
Fruit—Other	−0.23	**0.41**	−0.24	**−0.33**	0.03	**0.49**
Fruit—Citrus, melons, and berries	−0.20	0.18	−0.19	**−0.36**	0.04	**0.50**
Tomatoes	−0.21	0.04	−0.17	**−0.36**	**0.48**	0.14
Dark-Green Vegetables	−0.21	−0.19	−0.26	−0.22	0.26	**0.53**
Dark-Yellow Vegetables	−0.16	0.10	**−0.34**	−0.06	0.14	**0.45**
Other Vegetables	**−0.50**	0.17	**−0.65**	**−0.48**	**0.46**	**0.45**
Potatoes	**0.41**	0.25	0.06	0.05	0.16	−0.04
Other Starchy Vegetables	−0.03	0.16	−0.11	−0.15	−0.12	0.17
Legumes	0.01	−0.24	0.21	0.23	0.04	−0.08
Soy	−0.08	−0.11	−0.20	0.22	0.08	0.21
Refined Grains	−0.13	−0.12	0.17	0.13	0.11	**−0.34**
Whole Grains	−0.20	**0.47**	−0.25	−0.27	−0.05	**0.38**
Nuts	−0.28	0.10	**−0.31**	−0.02	0.18	0.19
Added Sugars	**0.76**	−0.28	**0.64**	**0.48**	**−0.32**	−0.27
FI	--					
Age	−0.28	--				
SNAP	0.80	−0.37	--			
Household Size	0.63	−0.50	0.62	--		
Modified Western	−0.26	0.09	−0.29	−0.31	--	
Prudent	−0.40	0.35	−0.56	−0.41	0.16	--

† Dietary pattern obtained using penalized logistic regression. ‡ Dietary pattern obtained using principal components analysis (PCA). Correlation coefficients (*r*) ≥ |0.30| are bolded to ease the identification of notable food groups characterizing the different patterns.

**Table 3 ijerph-19-14062-t003:** Derived dietary patterns and their distributions across study covariates in cancer survivors (subsample A in Figure 1) included in the survival analysis (n = 3317).

	Food Insecurity (FI) Pattern ^†^		Age Pattern ^†^		Food Assistance (SNAP) Pattern ^†^		Household Size Pattern ^†^		Modified Western ^‡^		Prudent ^‡^	
	M1 (*n* = 1658)	M2 (*n* = 1659)	M1 (*n* = 1658)	M2 (*n* = 1659)	M1 (*n* = 1658)	M2 (*n* = 1659)	M1 (*n* = 1658)	M2 (*n* = 1659)	M1 (*n* = 1658)	M2 (*n* = 1659)	M1 (*n* = 1658)	M2 (*n* = 1659)
Age		**		**		**		**				**
Mean (SD)	64.1 (13.5)	61.1 (15.9)	58.6 (14.9)	67.4 (13.1)	64.5 (13.3)	60.5 (16.0)	65.2 (13.3)	59.8 (15.8)	63 (15.1)	62.3 (14.5)	60.3 (15.4)	65.1 (13.6)
Sex										*		**
Male	752 (41.0)	775 (40.8)	741 (41.1)	786 (40.6)	762 (41.1)	765 (40.6)	741 (38.9)	786 (43.0)	821 (44.1)	706 (38.2)	836 (46.9)	691 (34.4)
Female	906 (59.0)	884 (59.2)	917 (58.9)	873 (59.4)	896 (58.9)	894 (59.4)	917 (61.1)	873 (57.0)	837 (55.9)	953 (61.8)	822 (53.1)	968 (65.6)
Race/Ethnicity		*		**		**		**		**		*
Minority	513 (14.2)	585 (17.2)	663 (18.2)	435 (12.6)	474 (12.8)	624 (18.8)	428 (11.8)	670 (19.8)	639 (19.3)	459 (12.6)	528 (14.2)	570 (17.3)
White	1145 (85.8)	1074 (82.8)	995 (81.8)	1224 (87.4)	1184 (87.2)	1035 (81.2)	1230 (88.2)	989 (80.2)	1019 (80.7)	1200 (87.4)	1130 (85.8)	1089 (82.7)
Education		**		*		**		**				**
≤HS	691 (31.5)	886 (42.6)	752 (34.4)	825 (39.6)	700 (30.3)	877 (43.9)	722 (33.3)	855 (40.6)	800 (37.6)	777 (36.2)	867 (40.2)	710 (33.2)
≥Some College	966 (68.5)	771 (57.4)	905 (65.6)	832 (60.4)	957 (69.7)	780 (56.1)	934 (66.7)	803 (59.4)	856 (62.4)	881 (63.8)	791 (59.8)	946 (66.8)
FIPR		**				**		**		**		*
≥1.3	1214 (86.9)	1065 (77.2)	1104 (81.4)	1175 (83.2)	1232 (87.5)	1047 (76.4)	1189 (84.9)	1090 (79.4)	1100 (79.2)	1179 (84.8)	1089 (80.2)	1190 (84.5)
<1.3	327 (13.1)	473 (22.8)	443 (18.6)	357 (16.8)	307 (12.5)	493 (23.6)	342 (15.1)	458 (20.6)	437 (20.8)	363 (15.2)	462 (19.8)	338 (15.5)
Household size												
<5 Persons	1547 (94.5)	1480 (90.0)	1460 (90.0)	1567 (95.1)	1563 (95.4)	1464 (89.0)	1557 (96.6)	1470 (87.7)	1495 (91.8)	1532 (92.8)	1482 (90.4)	1545 (94.4)
≥5 Persons	111 (5.5)	179 (10.0)	198 (10.0)	92 (4.9)	95 (4.6)	195 (11.0)	101 (3.4)	189 (12.3)	163 (8.2)	127 (7.2)	176 (9.6)	114 (5.6)
BMI												
Mean (SD)	29.3 (6.6)	29 (6.6)	29.3 (6.8)	29 (6.4)	29 (6.3)	29.4 (6.9)	29.2 (6.3)	29.1 (6.9)	28.6 (6.2)	29.7 (6.9)	29.7 (6.9)	28.6 (6.2)
Weekly MET Minutes				*								
Mean (SD)	2185.6 (3865.4)	2454.4 (5054.7)	2611.5 (4911.5)	1959 (3862.2)	2117.8 (3584.0)	2529.2 (5274.0)	2056.1 (3644.8)	2593.5 (5214.2)	2313.3 (4364.1)	2314.9 (4567.9)	2504.3 (5077.2)	2108.2 (3705.2)
Daily Caloric Intake												
Mean (SD)	1836.2 (660.8)	1958.3 (709.2)	1938.4 (696.4)	1842.8 (672.3)	1842.1 (655.0)	1952.2 (716.2)	1848 (655.0)	1945.1 (716.8)	1898.6 (705.1)	1891.3 (671.6)	2041.2 (746.9)	1736.6 (575.7)
CCI												
Mean (SD)	3.1 (1.4)	3.0 (1.4)	2.9 (1.3)	3.1 (1.5)	3.0 (1.4)	3.0 (1.4)	3.0 (1.4)	3.0 (1.4)	3.0 (1.4)	3.0 (1.4)	3.0 (1.4)	3.0 (1.4)
Food Security												
Food Secure	1482 (93.7)	1357 (83.1)	1351 (85.4)	1488 (92.5)	1512 (95.0)	1327 (81.7)	1476 (92.9)	1363 (84.0)	1379 (86.5)	1460 (90.5)	1352 (84.7)	1487 (92.9)
Food Insecure	176 (6.3)	302 (16.9)	307 (14.6)	171 (7.5)	146 (5.0)	332 (18.3)	182 (7.1)	296 (16.0)	279 (13.5)	199 (9.5)	306 (15.3)	172 (7.1)
SNAP Assistance				*								
No	1482 (93.7)	1357 (83.1)	1351 (85.4)	1488 (92.5)	1512 (95.0)	1327 (81.7)	1476 (92.9)	1363 (84.0)	1379 (86.5)	1460 (90.5)	1352 (84.7)	1487 (92.9)
Yes	176 (6.3)	302 (16.9)	307 (14.6)	171 (7.5)	146 (5.0)	332 (18.3)	182 (7.1)	296 (16.0)	279 (13.5)	199 (9.5)	306 (15.3)	172 (7.1)
Smoking Status						*						**
Current	188 (11.4)	329 (21.9)	344 (20.6)	173 (11.4)	191 (11.4)	326 (21.9)	179 (11.3)	338 (21.9)	284 (18.3)	233 (14.8)	393 (24.5)	124 (7.6)
Former	724 (42.3)	623 (35.2)	628 (36.3)	719 (42.0)	712 (42.6)	635 (34.8)	727 (43.0)	620 (34.4)	664 (38.6)	683 (39.2)	670 (37.8)	677 (40.1)
Never	745 (46.3)	706 (43.0)	685 (43.1)	766 (46.6)	754 (46.0)	697 (43.3)	750 (45.7)	701 (43.6)	709 (43.2)	742 (46.0)	594 (37.7)	857 (52.2)
Alcohol Use				*								
Heavy	214 (16.3)	109 (9.1)	242 (18.9)	81 (5.6)	265 (20.7)	58 (4.2)	206 (16.1)	117 (9.3)	227 (19.9)	96 (6.9)	207 (15.2)	116 (10.2)
Moderate	272 (16.3)	226 (15.9)	264 (17.2)	234 (14.8)	294 (17.3)	204 (14.8)	266 (15.0)	232 (17.3)	276 (17.4)	222 (15.0)	250 (16.9)	248 (15.3)
Non-drinking	1172 (67.5)	1324 (75.0)	1152 (63.9)	1344 (79.6)	1099 (62.0)	1397 (81.0)	1186 (68.9)	1310 (73.5)	1155 (62.7)	1341 (78.1)	1201 (67.9)	1295 (74.5)

** *p* < 0.01; * *p* < 0.05; M1 refers to the lower 50% fraction of the data while M2 refers to the upper 50% fraction of the data after splitting the diet scores at the median. **^†^** Dietary pattern obtained using penalized logistic regression. **^‡^** Dietary pattern obtained using principal components analysis (PCA).

**Table 4 ijerph-19-14062-t004:** Odds ratios and 95% confidence intervals for the relationship between the dietary patterns scores and the odds of being food insecure. There were 3317 cancer survivors (subsample A in Figure 1) that contributed to this analysis.

Dietary Pattern ^a^	Q1	Q2	Q3	Q4	Q5	*p* _Q5–Q1_	*p* _trend_	HR ^b^_continuous_	*p* ^c^ * _quadratic_ *
Food Insecurity ^†^	1.00	1.09 (0.58–2.02)	1.18 (0.57–2.45)	1.91 (1.04–3.53) *	2.42 (1.21–4.82) *	0.01 *	<0.01 **	1.50 (1.19–1.90) **	0.11
Age ^†^	1.00	1.91 (1.12–3.27) *	1.41 (0.67–2.93)	1.14 (0.49–2.69)	1.82 (0.93–3.56)	0.08	0.28	1.05 (0.87–1.27)	0.57
Food Assistance (SNAP) ^†^	1.00	1.38 (0.65–2.93)	1.44 (0.77–2.71)	2.54 (1.22–5.30) *	2.23 (1.26–3.94) **	<0.01 **	<0.01 **	1.37 (1.12–1.68) **	0.46
Household Size ^†^	1.00	1.63 (0.78–3.43)	1.00 (0.52–1.92)	2.77 (1.46–5.25) **	2.02 (0.98–4.18)	0.06	0.01 *	1.27 (1.04–1.54) *	0.36
Modified Western ^‡^	1.00	0.86 (0.48–1.51)	0.69 (0.33–1.45)	1.46 (0.81–2.64)	1.33 (0.66–2.67)	0.42	0.16	1.05 (0.88–1.25)	0.70
Prudent ^‡^	1.00	0.81 (0.37–1.78)	1.09 (0.53–2.26)	0.54 (0.26–1.10)	0.40 (0.20–0.80) **	<0.01 **	<0.01 **	0.76 (0.63–0.92) **	0.27

** *p* < 0.01; * *p* < 0.05; ^a^ All models adjusted for age, sex, race/ethnicity, family income-to-poverty ratio, highest level of education attained, household size, SNAP participation status, BMI, estimated caloric intake, weekly MET minutes, primary cancer site, smoking status, and the Charlson Comorbidity Index score and were weighted according to guidelines provided by the NCHS. ^b^ Hazard ratio (HR) corresponding to a standard deviation increase in the diet pattern score. ^c^ Wald test *p*-value for a quadratic polynomial term. **^†^** Dietary pattern obtained using penalized logistic regression. **^‡^** Dietary pattern obtained using principal components analysis (PCA).

## Data Availability

All data used in the analyses are publicly available from the Centers for Disease Control and Prevention (https://wwwn.cdc.gov/nchs/nhanes/Default.aspx) (accessed on 24 October 2022). Additionally, R code and data used specifically in these analyses are also available in the following GitHub repository: https://github.com/cmainov/NHANES-Diet-Penalized-Regression (accessed on 24 October 2022).

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
