# Peer review of "Empirical Dietary Patterns Associated with Food Insecurity in U.S. Cancer Survivors: NHANES 1999–2018"

_ijerph, 2022, doi:10.3390/ijerph192114062_

Round 1

Author Response

We thank reviewer 1 for taking the time to carefully review our manuscript and provide meaningful commentary. Below are point-by-point responses to their comments:

Comment #1: Abstract:
The abstract would benefit from a more detailed description of the results, particularly
regarding which dietary intake patterns were strongly associated with FI.

Response #1: We thank the reviewer for this suggestion. We have updated the abstract to include some commentary about the specific patterns that had strong associations with food insecurity.

Comment #2: Please explicitly mention whether similar studies have been performed. If so, what are the differences between the present NHANES study and those studies?

Response #2: We thank the reviewer for this suggestion. We have added some language at the end of the introduction section to address this.

Comment #3: Please provide a reference to support the claim that “These evidence-based guidelines may improve QOL and disease outcomes”

Response #3: We thank the reviewer for noting this. We have updated the citations for this sentence and added a few references to support the claim.

Comment #4: “as these foods may bolster cancer risk and progression” change to “as higher intakes of these foods may be associated with an increased cancer risk and worsen the prognosis.”

Response #4: we thank the reviewer for this suggestion. We have updated the language accordingly.

Comment #5: Please describe in the text why individuals reporting a history of non-melanoma skin cancer were recorded as not having a significant cancer history.

Response #5: We thank the reviewer for this suggestion. We have updated the language in this section to make it clear why these individuals were recoded.

Comment #6: Please describe in the text whether the physical activity domain includes occupational and recreational physical activity.

Response #6: We thank the reviewer for this suggestion. The NHANES PAQ module does not occupation-specific questions and, instead, aims to capture/measure daily physical activity which, conceivably, includes occupational physical activity. (see: https://wwwn.cdc.gov/Nchs/Nhanes/2017-2018/PAQ_J.htm). We have, nevertheless, updated the language in this section of the manuscript to make it clear that the PAQ accounts for all physical activity.

Comment #7: It would be helpful for the readers if the authors could list the 10 items for households without children and eight for households with children.

Response #7: We thank the reviewer for this suggestion. We have updated the references in this section to include a reference that takes readers to a PDF of the survey so that they can view it.

Reviewer 2 Report

I really enjoyed reading this thorough, with a very good flow. thank you. Minor comments:

-  You have specified that the participants provided consent. Can you specify whether written or oral consent? Also you referenced 14- checking up this ref, it looks a kind of note instead of citation. Suggest moving it into the text.

-           You have specified “Alcohol consumption was not included as a covariate in these models to minimize collinearity, given that the extracted patterns already considered alcohol consumption in their computation..” (Line 202-204). I haven’t read this in the top part of the manuscript. was the extraction done by before accessing the data or by the researchers? If earlier-can you provide a reference (can be a guide????)

-          On Table 2 (Line 283) you stated that subsample A, an analytical subsample of food insecure cancer survivors. As per Figure 1, it looks like only 433 are food insecure. Can you double check this.

-          I can’t see title for Tables 3 and 4. Coming back- I think the titles are written under the tables- please check.

-          Assuming Table 3 for line 315, you have defined M1 and M2 (footnote), can you be clear what was the parameter for this (randomly dividing the sample?)

-          Interpreting PCA (line 287) – you have specified only the first and second components were interpretable – can you please include in the results the logic behind with reference included? It is <0.3 usually considered a cut of point – please check and provide evidence.

Author Response

We thank reviewer 1 for taking the time to carefully review our manuscript and provide meaningful commentary. Below are point-by-point responses to their comments:

Comment #1: You have specified that the participants provided consent. Can you specify whether written or oral consent? Also you referenced 14- checking up this ref, it looks a kind of note instead of citation. Suggest moving it into the text.

Response #1: We thank the reviewer for this suggestion. We have updated the text to reflect “written” consent. Moreover, we thank the reviewer for catching this mistake in the references. That reference has been updated.

Comment #2: You have specified “Alcohol consumption was not included as a covariate in these models to minimize collinearity, given that the extracted patterns already considered alcohol consumption in their computation..” (Line 202-204). I haven’t read this in the top part of the manuscript. was the extraction done by before accessing the data or by the researchers? If earlier-can you provide a reference (can be a guide????)

Response #2: We thank the reviewer for sharing this concern. We have updated the sentence in the introduction delineating the goal of the study to make it clear that we, the investigators, are performing dietary patterns extraction on the data and that the data do not come with the dietary patterns.

Comment #3: On Table 2 (Line 283) you stated that subsample A, an analytical subsample of food insecure cancer survivors. As per Figure 1, it looks like only 433 are food insecure. Can you double check this.

Response #3: We thank the reviewer for catching this typo. It has been corrected.

Comment #4:  I can’t see title for Tables 3 and 4. Coming back- I think the titles are written under the tables- please check.

Response #4: Thank you for catching these formatting issues. We have moved the titles to the top of the tables for tables 3 and 4.

Comment #5: Assuming Table 3 for line 315, you have defined M1 and M2 (footnote), can you be clear what was the parameter for this (randomly dividing the sample?)

Response #5: Thank you for this suggestion. We have updated the language to make clear that M1 and M2 represent the lower and upper fractions of the data after a median split.

Comment #6: Interpreting PCA (line 287) – you have specified only the first and second components were interpretable – can you please include in the results the logic behind with reference included? It is <0.3 usually considered a cut of point – please check and provide evidence.

Response #6: we thank the reviewer for this thoughtful comment. We have updated language in this section and added a new reference. We are unclear if the reviewer is referring to the procedure for identifying which components to retain or if they refer to identifying food groups that substantially contribute to the different dietary patterns. We will address both. It is commonplace to use a scree plot and retain all of the components up to the “elbow”. Moreover, interpretability is a big factor and several other studies of nutritional epidemiology use interpretability of the components as a guide for the number of components to retain. There are arbitrary values/cut points (such as Kaiser’s Rule) that can be used but we feel that the approach that we implemented (which is still commonplace in the multivariate statistics literature) is sound and has been used routinely by other nutritional epidemiology studies. Concerning, the identification of food groups that contribute to a dietary pattern, there is an arbitrary rule for identifying a significant contribution from a food group if the factor loading or correlation is greater than or equal to the absolute value of 0.30. To make this point clear, we have bolded entries in Table 2 that have an absolute value of greater than or equal to 0.30. We also changed the language in the paragraph discussing Table 2 (just prior to the Table) to only point out food groups that satisfy this criteria of greater than or equal to 0.30. Moreover, we add a reference for this in Table 2.

Reviewer 3 Report

Dear authors,

Thank you for writing this interesting article.

However, there are some comments needed to be answered:

-          Why did you not evaluate the dietary intake using FFQ? A 24h recall might result in bias in final analyses. How did you manage that!

-          You used an average of two 24h recalls. What was the exact time of these 2 days? How did you manage the gap between the days and the probable differences regarding the food intakes between the two 24h recalls.

-          As you stated, previous studies including a study by Trego et al. found that a low proportion of cancer survivors indicated being food insecure, however they did not find a significant moderating effect of time since diagnosis on food insecurity -related outcomes. So please bring this importance (“time since diagnosis on food insecurity -related outcomes”) in abstract as well as the introduction part as a novelty of your work.

-          Why did you not consider a family history of cancer as a covariate in you analysis?

-          Although considering the dietary intake as a covariate is acceptable, I was wondering that why did you not adjust its intake before the analyses?

-          Were there any differences among participants regarding the ability of access to food which would affect the status of food security (for example due to different locations)? Please consider this importance.

-          Only 13% of total participants were considered to have food insecurity. As a result, the related findings on this subsample would be interpreted with caution (please also consider this in the abstract).

-          Self-reporting food insecurity would also be another limitation of your study. Please discuss.

Author Response

Comment #1: Why did you not evaluate the dietary intake using FFQ? A 24h recall might result in bias in final analyses. How did you manage that!

Response #1: We thank the reviewer for this consideration. Ideally, having an FFQ is a better approach than a 24-hr recall as it is intended to capture long-term intake. However, the NHANES datasets contain data for 24-hr recall. A small subset of cycles –not all used in this analysis—use an FFQ. However, to include as many observations as possible and to also have observations from more recent cycles represented, the best course of action was to use 24-hr recall data from NHANES. We have updated the limitations section of the manuscript to include the use of a 24-hr recall as opposed to a FFQ as a limitation of the study.

Comment #2: You used an average of two 24h recalls. What was the exact time of these 2 days? How did you manage the gap between the days and the probable differences regarding the food intakes between the two 24h recalls.

Response #2: We thank the reviewer for this thoughtful question. The exact protocols for how the 24-hr recalls were administered are publicly available on the NHANES website. In short, the NHANES protocol involves a first 24-hr recall that is administered and then a subsequent one that is randomly administered 3-10 days afterwards, in an attempt to control for certain biases. Having a single day of dietary data to work with may introduce bias as we know that diet varies across days of the week. Therefore, including two days of data is expected to minimize such bias. Finally, we provided two citations (citations 23, and 24) that have used this approach previously and we discuss the general protocol for administering the 24-hr recalls in our methods section (Section 2.3 Dietary Assessment Data).

Comment #3: As you stated, previous studies including a study by Trego et al. found that a low proportion of cancer survivors indicated being food insecure, however they did not find a significant moderating effect of time since diagnosis on food insecurity-related outcomes. So please bring this importance (“time since diagnosis on food insecurity-related outcomes”) in abstract as well as the introduction part as a novelty of your work.

Response #3: We thank the reviewer for this suggestion and praise. We have updated the abstract and introduction accordingly.

Comment #4: Why did you not consider a family history of cancer as a covariate in you analysis?

Response #4: The NHANES Medical Conditions questionnaire (MCQ) does not probe subjects on their family history of cancer, only on personal history. Thus, we are limited by this methodological constraint. Otherwise, it is a conceivable hypothesis that a family history of cancer may exacerbate food insecurity issues in cancer survivors. However, this was outside the scope of this analysis.

Comment #5: Although considering the dietary intake as a covariate is acceptable, I was wondering that why did you not adjust its intake before the analyses?

Response #5: We thank the reviewer for this thoughtful comment. We believe the reviewer is referring to energy adjustment. Our analysis did adjust for caloric intake using a multivariate density model (i.e, all dietary intake values were divided by the subject’s estimated caloric intake and then the final models included estimated caloric intake as a covariate to adjust the model with—see Willet et al. ).

Comment #6: Were there any differences among participants regarding the ability of access to food which would affect the status of food security (for example due to different locations)? Please consider this importance.

Response #6: We thank the reviewer for this thoughtful comment. This is an excellent observation. However, geographic data/geocodes of the participants are not available publicly. There is a restricted access file that NHANES provides with geographic data, but it remains to be seen if the type of data in those files can be used to extract some sort of data on proximity to food deserts or other locations that may hamper a subject’s access to outlets that provide healthy and nutritious foods (no studies to our knowledge have used those data in this capacity).

Comment #7: Only 13% of total participants were considered to have food insecurity. As a result, the related findings on this subsample would be interpreted with caution (please also consider this in the abstract).

Response #7: we thank the reviewer for this comment. We remind the reviewer that NHANES provides nationally representative data for the US. Through cluster, stratified sampling, and other probabilistic methods, a sample is generated from the US civilian non-institutionalized population. Given that we know the prevalence of food insecurity is ~10.8% across the US, it is not far-fetched to consider that the prevalence of food insecurity amongst cancer survivors is around 13% (as you and our unweighted results suggest). We would like to make clear that our conclusions are only generalizable to the food insecure cancer-survivor population in the United States and not any other group outside of that. The language in the conclusions section has been carefully written to make this point clear. The prevalence in our study was quite similar to those figures reported elsewhere (such as by Trego et al.).

Comment #8: Self-reporting food insecurity would also be another limitation of your study. Please discuss.

Response #8: We agree with the reviewer on this comment and have updated the limitations section of the discussion accordingly.

Reviewer 4 Report

The titles of the Table 3, Table 4 and supplementary tables should be on top.

Supplementary Figure 1 is the most important figure and should be included in the main text, not in the supplementary. The reason is that it showed the results of penalized logistic regression, which is the key methodology and base of this article.

According to the results of penalized logistic regression, what suggestions of dietary pattern will the authors provide to cancer patients and healthy individuals? For example, what is the suggested ranges of the consumption amounts of major food/ingredient groups? The consumption frequency of eating certain types of foods?

What is the major impact of the research, scientifically and societally?

How this study contributed to the wellbeing not only limited to US but to solve the problem global level?

Some of the words, especially some terms, are uncommonly used words. The aim should make the article easy to understand.

Author Response

Comment #1: The titles of the Table 3, Table 4 and supplementary tables should be on top.

Response #1: We thank the reviewer for catching this. These titles have been moved accordingly.

Comment #2: Supplementary Figure 1 is the most important figure and should be included in the main text, not in the supplementary. The reason is that it showed the results of penalized logistic regression, which is the key methodology and base of this article.

Response #2: We thank the reviewer for this suggestion. We have incorporated this figure into the main text as Figure 2.

Comment #3: According to the results of penalized logistic regression, what suggestions of dietary pattern will the authors provide to cancer patients and healthy individuals? For example, what is the suggested ranges of the consumption amounts of major food/ingredient groups? The consumption frequency of eating certain types of foods?

Response #3: We thank the reviewer for these thoughtful comments. As we emphasize in our introduction and discussion sections, cancer survivors should be adhering as closely as possible to the guidelines put forth by the American Institute for Cancer Research, which are comprehensive and evidence-based recommendations for those living with cancer and healthy individuals who are interested in preventing cancer. Moreover, a posteriori (or data-driven) methods for deriving dietary patterns have a limitation in that they only reveal associations and not absolute values of intake for any food group. This is in contrast to a priori (or guideline-based) diet quality indices where there might be absolute intake values that are relied upon for the calculation of those indices. Thus, given the results we have generated, we would be extrapolating from those results if we were to make explicit recommendations based on dietary intake cut-offs. However, we have updated the limitations piece in the discussion section to make this point clear and also added a sentence emphasizing the need to use the AICR guidelines clinically. This research is showing that food insecure cancer survivors are not meeting these recommendations and that they should be following the AICR recommendations at this time.  Finally, a key point is that the goal of this analysis was not to make recommendations but to analyze general patterns in the food insecure cancer survivor population of the United States.

Comment #4: What is the major impact of the research, scientifically and societally?

Response #4: We thank the reviewer for this probing question. We have detailed a number of ramifications of the research in our discussion section already. We have added a few more sentences to the discussion section and reworded some points to address the reviewers concern.

Comment #5: How this study contributed to the wellbeing not only limited to US but to solve the problem global level?

Response #5: We thank the reviewer for this consideration. We have added some language to the manuscript talking about how the methodology we implemented can be used in a variety of population contexts as well as for the purposes of tracking policy benchmarks (which can also be used in a global context). We also added some language about how these findings are important in a global context given similarities and dissimilarities between cancer survivors in the U.S. and abroad.

Comment #6: Some of the words, especially some terms, are uncommonly used words. The aim should make the article easy to understand.

Response #6: We thank the reviewer for this comment. We cannot assume to what specific elements of language the reviewer is referring to. However, we have gone through the entire manuscript and altered what we believe may be some sections or terms that are not completely understandable. If there are other words that the reviewer would like us to remove or change, please let us know, specifically, which terms are problematic.

Round 2

Reviewer 3 Report

Dear authors, 

Thank you for your responses. the manuscript has been improved. 

Reviewer 4 Report

The authors have modified the manuscript corresponding to the comments.